# Analysis of the Threshold for Energy Consumption in Displacement of Random Sensors

**DOI:** 10.3390/s22228789

**Published:** 2022-11-14

**Authors:** Rafał Kapelko

**Affiliations:** Department of Fundamentals of Computer Science, Wrocław University of Science and Technology, 50-370 Wrocław, Poland; rafal.kapelko@pwr.edu.pl

**Keywords:** coverage, interference, random, displacement, energy, sensors, Beta distribution

## Abstract

The fundamental problem of energy-efficient reallocation of mobile random sensors to provide full coverage without interference is addressed in this paper. We consider *n* mobile sensors with the identical sensing range placed randomly on the unit interval and on the unit square. The sensors move from their initial random positions to the final locations so that: (a) every point on the unit interval or on the unit square is within the range of a sensor; (b) each pair of sensors is at a Euclidean distance greater than or equal to *s*; (c) the energy consumption for the movement of the sensors to the final positions is minimized. As a cost measure for the energy in the movement of sensors, we consider *a*-total movement defined as the sum $\sum_{i=1}^{n}d_{i}^{a}$, for some constant a>0, provided that the *i*-th sensor is displaced the distance $d_{i}$. The main contribution is summarized as follows: (1) if the sensors are placed on the unit interval, we explain the sharp increase around the sensing radius equal to $\frac{1}{2n}$ and the interference distance equal to $\frac{1}{n}$ for the expected minimal *a*-total displacement; (2) if the sensors are placed on the unit square, we explain the sharp increase around the square sensing radius equal to $\frac{1}{2\sqrt{n}}$ and the interference distance equal to $\frac{1}{\sqrt{n}}$ for the expected minimal *a*-total displacement. We designed and analysed three algorithms. The probabilistic analysis of our protocols is based on a novel mathematical theory of the Beta distribution.

## 1. Introduction

Mobile sensors are being deployed in many application areas to enable easier information retrieval in communication environments, from sensing and diagnostics to critical infrastructure monitoring (e.g., see [1,2,3]).

The current reduction in manufacturing costs makes random deployment of the sensors more attractive. Since existing sensor deployment scenarios cannot always ensure precise placement of sensors, their initial deployment may be somewhat random. In some cases, the sensors may have drifted to new arbitrary positions over time. Even initially deterministically placed sensors may create random patterns of effectiveness due to failures.

A typical sensor is able to sense and, thus, cover a bounded region specified by its sensing radius [4]. To monitor and protect a larger region against intruders, every point of the region has to be within the sensing range of a sensor. It is also known that proximity between sensors affects the transmission and reception of signals and causes the degradation of performance [5]. Therefore, in order to avoid interference, a critical value, say *s*, is established. It is assumed that, for a given parameter *s*, two sensors interfere with each other during communication if their distance is less than *s* (see [6,7]). However, random deployment of the sensors might leave some gaps in the coverage of the area, and the sensors may be too close to each other. Therefore, to attain coverage of the area and to avoid interference, the reallocation of sensors may be the only option. Moreover, the ability to move the mobile sensors to the final destinations is not unrealistic. Clearly, the displacement of a team of sensors should be performed in the most efficient way.

The energy consumptionfor the displacement of a set of *n* sensors is measured by the sum of the respective displacements and the power of the individual sensors. We define below the concept of *a*-total displacement.

**Definition** **1** (*a*-total displacement).
*Let a>0 be a constant. Suppose the displacement of the i-th sensor is a distance $d_{i}$. The a-total displacement is defined as the sum $\sum_{i=1}^{n}d_{i}^{a}$.*


The motivation for this cost metric arises from the fact that the parameter *a* in the exponents represents various conditions on the region lubrication and friction, which affect the sensor movement.

We consider *n* mobile sensors, which are placed independently and uniformly at random on the unit interval and on the unit square.

For the case of unit interval [0,1], each sensor is equipped with an omnidirectional antenna of identical sensing radius $r_{1}>0$. Thus, a sensor placed at location *x* on the unit interval can cover any point at a distance at most $r_{1}$, either to the left or the right of *x* (see Figure 1a).

For the case of unit square ${\left[0,1\right]}^{2}$, each sensor has the identical square sensing radius $r_{2}>0$.

**Definition** **2** (cf. [8] square sensing radius)**.**
*We assume that a sensor located in position $\left(x_{1},x_{2}\right)$ where $0\leq x_{1},\;x_{2}\leq 1$ can cover any point in the area delimited by the square with corner points $\left(x_{1}\pm r_{2},\;x_{2}\pm r_{2}\right)$ and call $r_{2}$ the square sensing radius of the sensor.*

The concept of the square sensing radius was introduced in the paper [8]. Figure 1b illustrates the square sensing radius.

However, in most cases, the sensing area of a sensor is a circular disk of radius $r_{c}$, but our upper bound result, proven in the sequel, for square sensing radius $r_{2}$ is obviously valid for a circular disk of radius $r_{c}$ equal to $\sqrt{2}r_{2}$ circumscribing the square.

The sensors are required to move from their current random locations (see Figure 2) to new positions to satisfy the following requirement.

**Definition** **3**($\left(r_{m},s\right)$-C&I requirement)**.**
*Fix m∈{1,2}. A set of sensors placed on the m-dimensional unit cube satisfies the $\left(r_{m},s\right)$ coverage and interference requirements:*
(*a*) *Every point on the m-dimensional unit cube ${\left[0,1\right]}^{m}$ is within the range $r_{m}$ of a sensor, i.e., the m-dimensional unit cube is completely covered.*(*b*) *Each pair of sensors is placed at a Euclidean distance greater than or equal to s.*

In this paper, we investigate the problem of energy-efficient displacement of the random mobile sensors.

**Definition** **4** (Energy efficient displacement)**.**
*Assume that n mobile sensors are placed independently and uniformly at random on the unit interval or on the unit square. The sensors move from their initial random location to the final destination so that, in their final placement, the sensor system satisfies the $\left(r_{m},s\right)$-coverage and interference requirement and the a-total displacement is minimized in expectation.*


In WSNs, energy consumption is the fundamental problem to study. It is known that the sensors consume much more energy during movement than during sensing or communication [9]. The proposed solution can be widely used in border surveillance to detect intruders illegally crossing the protected area.

Throughout the paper, we use the Landau asymptotic notations:(i)f(n)=O(g(n)) if there exist a constant $C_{1}>0$ and integer *N* such that $\left|f\left(n\right)\right|\leq C_{1}\left|g\left(n\right)\right|$ for all n>N;(ii)f(n)=Ω(g(n)) if there exist a constant $C_{2}>0$ and integer *N* such that $\left|f\left(n\right)\right|\geq C_{2}\left|g\left(n\right)\right|$ for all n>N;(iii)f(n)=Θ(g(n)) if and only if f(n)=O(g(n)) and f(n)=Ω(g(n)).

### 1.1. Contribution and Outline of the Paper

Let a>0 be a constant. Assume that *n* mobile sensors with the identical sensing radius $r_{1}$ and square sensing radius $r_{2}$ are placed independently at random with the uniform distribution on the unit interval and on the unit square.

In this paper, we give the picture of the threshold phenomena for the coverage and interference requirement in one dimension, as well as in two dimension (see Definition 3). The *a*-total displacement (the energy consumption) is used to measure the movement cost (see Definition 1), while the Euclidean distance is used for the interference distance, and the sensing area of a sensor in two dimension is a square (see Definition 2). Let us also recall that, in two dimension, the sensors can move directly to the final locations via the shortest route, not only in a vertical and horizontal fashion.

Let ϵ>0, 1>δ>0 be arbitrary small constants independent on the number of sensors *n*.

Table 1 summarizes our main contribution in one dimension.

We prove the following results.

(1)When the sensing radius $r_{1}=\frac{1}{2n}$ and the interference distance $s=\frac{1}{n}$, the expected minimal *a*-total displacement for the $(r_{1},s)-C\&I$ requirement is in $\Theta \left(n^{\frac{a}{2}}n^{1-a}\right)$.(2)As the sensing radius $r_{1}=\frac{1+\epsilon }{2n}$ is a little above $\frac{1}{2n}$ and the interference distance $s=\frac{1-\delta }{n}$ is a little below $\frac{1}{n}$, the expected minimal *a*-total displacement for the $(r_{1},s)-C\&I$ requirement sharply declines to $O\left(n^{1-a}\right)$.

Table 2 summarizes our main contribution in two dimensions.

We prove the following results.

(1)When the square sensing radius $r_{2}=\frac{1}{2\sqrt{n}}$ and the interference distance $s=\frac{1}{\sqrt{n}}$, the expected minimal *a*-total displacement for the $(r_{2},s)-C\&I$ requirement is in $\Omega \left({\left(\ln(n)\right)}^{\frac{a}{2}}n^{1-\frac{a}{2}}\right)$.(2)As the square sensing radius $r_{2}=\frac{1+\epsilon }{2\lfloor \sqrt{n}\rfloor }$ is a little above $\frac{1}{2\sqrt{n}}$ and the interference distance $s=\frac{1-\delta }{\left\lfloor \sqrt{n}\right\rfloor }$ is a little below $\frac{1}{\sqrt{n}}$, the expected minimal *a*-total displacement for the $(r_{2},s)-C\&I$ requirement sharply declines to $O\left(n^{1-\frac{a}{2}}\right)$.

Notice that *n* sensors on the unit interval [0,1] with sensing radius $r_{1}=\frac{1}{2n}$ and the interference distance have to move to the anchor positions to satisfy the $\left(r_{1},s\right)$-coverage and interference. When $r_{1}>\frac{1}{2n}$ and $s<\frac{1}{n}$, there are no anchor positions predetermined in advance. A similar remark holds for the sensors on the unit square ${\left[0,1\right]}^{2}$.

Our theoretical results imply that the expected *a*-total displacement is constant and independent of the number of sensors for some parameters *a*. Namely, we have the following upper bounds:(i)For the random sensors on the unit interval, when
(1)$$n(2r_{1})=1+\epsilon ,$$
i.e., the sum of the sensing area of *n* sensors is a little bigger than the length of the unit interval, it is possible to provide the full area coverage in O(1) expected *a*-total displacement with a≥1.(ii)For the random sensors on the unit square, when
(2)$$n{\left(2r_{2}\right)}^{2}∼{\left(1+\epsilon \right)}^{2}\;\;\;\;\;\mathrm{as}\;\;\;\;\;n\to \infty ,$$
i.e., the sum of the sensing area of *n* sensors is asymptotically a little bigger than the area of unit square, the expected *a*-total displacement with a≥2 to provide full area coverage is O(1). Obviously, this result is easily applicable to the model when the sensing area of a sensor is a circular disk of radius $r_{c}$ by taking the circle circumscribing the square. Namely, when
$$n\pi {\left(r_{c}\right)}^{2}∼\frac{\pi }{2}{\left(1+\epsilon \right)}^{2}\;\;\;\;\;\mathrm{as}\;\;\;\;\;n\to \infty$$
then the expected two-total displacement to provide full area coverage is constant.

This constant cost seems to be of practical importance due to efficient monitoring against illegal trespassers. It is well known that intrusion detection is an important application of wireless sensor networks. In this case, it is necessary to ensure coverage with good communication.

Notice that the constant expected cost in (i) and (ii) is valid for *n* random sensors with the identical sensing radius $r_{1}=\frac{x(1+\epsilon )}{2n}$ on the interval of length *x* and for *n* random sensors with the identical square sensing radius $r_{2}=\frac{x(1+\epsilon )}{2\lfloor \sqrt{n}\rfloor }$ on the square [0,x]×[0,x].

We also present three algorithms (see Algorithms 1–3). It is worthwhile to mention that, even though the algorithms are simple, the analysis is challenging. Notice that Algorithms 1–3 can be implemented by a centralized controller telling each sensor where and when to move. In Section 2, we prove some technical properties of the Beta distribution with the special positive integer parameters needed in the current paper (see Lemmas 2 and 3).

The overall organization of the paper is as follows. Section 1.2 briefly summarizes some related work. Section 2 gives some properties of the Beta distribution, the results of which are used to analyse the $\left(r_{m},s\right)$-C&I requirement in WSNs. Section 3 and Section 5 deal with sensors on the unit interval. In Section 4 and Section 6, we investigate sensors on the unit square, while further insights into the higher dimension are discussed in Section 7. Section 8 deals with the experimental evaluation of Algorithm 1. Section 9 contains conclusions and directions for future work. Finally, for the sake of readability, certain technical proofs are deferred to the Appendices (Appendix A, Appendix B, Appendix C, Appendix D, Appendix E, Appendix F and Appendix G).
**Algorithm 1**MV(n,ρ,s) moving sensors on [0,1]**Require:** The initial locations of *n* mobile sensors, placed uniformly and independently at random on the unit interval [0,1].**Ensure:** The final positions of the sensors such that:
(i)The distance between consecutive sensors is greater than or equal to *s* and less than or equal to ρ.(ii)The leftmost sensor is at a distance less than or equal to $\frac{\rho }{2}$ from the origin.**Initialization:**   Sort the initial locations of *n* sensors with respect to the origin of the interval, the location of sensors after sorting $X_{\left(1\right)}\leq X_{\left(2\right)}\leq \cdots \leq X_{\left(n\right)};$
 1:Let $X_{0}=0;$ 2:**for**i=1*n***do** 3:   **if** $X_{\left(i\right)}-X_{\left(i-1\right)}<s$ **then** 4:     move left to right the sensor $X_{\left(i\right)}$ to the new position $\min\left(s+X_{\left(i-1\right)},1\right);$ 5:   **else**
**if**
$X_{\left(i\right)}-X_{\left(i-1\right)}>\rho$
**then** 6:     move right to left the sensor $X_{\left(i\right)}$ to the new position $\rho +X_{\left(i-1\right)};$ 7:   **else** 8:     do nothing; 9:   **end if**10:**end for**11:**if**$X_{\left(1\right)}>\frac{1}{2}\rho$**then**12:   $z:=X_{\left(1\right)}-\frac{1}{2}\rho ;$13:   **for** i=1*n* **do**14:     move right to left the sensor $X_{\left(i\right)}$ to the new position $X_{\left(i\right)}-z;$15:   **end for**16:**end if**

**Algorithm 2**$CV_{1}\left(n,r_{1},s\right)$ for the $\left(r_{1},s\right)$-coverage and interference requirement on [0,1] when $r_{1}=\frac{1+\epsilon }{2n},\;\;\;$$s=\frac{1-\delta }{n}$ provided that ϵ>0 and 1>δ>0 are fixed and independent of n

**Require:** The initial locations of *n* mobile sensors with the identical sensing radius $r_{1}=\frac{1+\epsilon }{2n}$, placed uniformly and independently at random on the unit interval [0,1].
**Ensure:** The final positions of sensors the to satisfy the $\left(r_{1},s\right)$-coverage and interference requirement on the interval [0,1].
**Initialization:**   Apply Algorithm MV(n,ρ,s) for $\rho :=\frac{1+\frac{\epsilon }{2}}{n},\;\;\;$$s:=\frac{1-\delta }{n}$ and the random sensors $X_{1},X_{2},\cdots ,X_{n}.$ Let $Y_{1},Y_{2},\cdots ,Y_{n}$ be the location of *n* sensors $X_{\left(1\right)}\leq X_{\left(2\right)}\leq \cdots \leq X_{\left(n\right)}$ after Algorithm MV(n,ρ,s);
 1:**switch** () 2:**case A**$\left(Y_{n}\geq 1-r_{1}\right)$ 3:   do nothing; 4:**case B**$\left(Y_{n}\in \left(1-\frac{2}{n^{\frac{a}{a+1}}},1-r_{1}\right)\right)$ 5:   move the sensor $Y_{n}$ to the new position $1-r_{1},\;\;\;$i:=n−1; 6:   **while**
$Y_{i+1}-Y_{i}>2r_{1}$
**do** 7:     move the sensor $Y_{i}$ to the new position $1-r_{1}-\left(n-i\right)2r_{1},\;\;\;$i:=i−1; 8:   **end while** 9:**case C** $\left(Y_{n}\leq 1-\frac{2}{n^{\frac{a}{a+1}}}\right)$10:  **for**
i=1
**to**
*n*
**do**11:     move the sensor $Y_{i}$ to the position $\left(\frac{i}{n}-\frac{1}{2n}\right);$12:  **end for**13:**end switch**


**Algorithm 3**$CV_{2}\left(n,r_{2},s\right)$ for the $\left(r_{2},s\right)$-coverage and interference requirement on the ${\left[0,1\right]}^{2}$ when $r_{2}=\frac{1+\epsilon }{2\lfloor \sqrt{n}\rfloor }$ and $s=\frac{1-\delta }{\left\lfloor \sqrt{n}\right\rfloor }$ provided that ϵ>0 and 1>δ>0 are fixed and independent of n
**Require:** The initial locations of *n* mobile sensors with the identical square sensing radius $r_{2}=\frac{1+\epsilon }{2\lfloor \sqrt{n}\rfloor }$, placed uniformly and independently at random on the unit square ${\left[0,1\right]}^{2}.$
**Ensure:** The final positions of the sensors satisfying the $\left(r_{2},s\right)$-coverage and interference requirement on the square ${\left[0,1\right]}^{2}.$

**Initialization:   **
Choose ${\left\lfloor \sqrt{n}\right\rfloor }^{2}$ sensors at random;Sort the initial locations of sensors according to the second coordinate; let the sorted locations be $S_{1}=\left(x_{1},y_{1}\right),$
$S_{2}=\left(x_{2},y_{2}\right),\ldots$
$S_{n}=\left(x_{n},y_{n}\right),\;\;\;$$y_{1}\leq y_{2}\leq \cdots \leq y_{n};$

1:
**for**

j=1

**to**

$\left\lfloor \sqrt{n}\right\rfloor$

**do**
2:   **for**
i=1
**to**
$\left\lfloor \sqrt{n}\right\rfloor$
**do**3:     move sensor $S_{\left(j-1\right)\left\lfloor \sqrt{n}\right\rfloor +i}$ to position     $\left(x_{\left(j-1\right)\left\lfloor \sqrt{n}\right\rfloor +i},\frac{j}{\left\lfloor \sqrt{n}\right\rfloor }-\frac{1}{2\lfloor \sqrt{n}\rfloor }\right)$4:   **end for**5:
**end for**
6:**for**j=1**to**$\left\lfloor \sqrt{n}\right\rfloor$ **do**7: apply Algorithm $CV_{1}\left(n,r_{1},s\right)$ for $n:=\lfloor \sqrt{n}\rfloor ,$$s:=\frac{1-\delta }{\left\lfloor \sqrt{n}\right\rfloor },$$r_{1}:=\frac{1+\epsilon }{2\lfloor \sqrt{n}\rfloor }$ and sensors $S_{\left(j-1\right)\left\lfloor \sqrt{n}\right\rfloor +1},S_{\left(j-1\right)\left\lfloor \sqrt{n}\right\rfloor +2},\cdots S_{\left(j-1\right)\left\lfloor \sqrt{n}\right\rfloor +\left\lfloor \sqrt{n}\right\rfloor };$8:
**end for**



### 1.2. Related Work

There are extensive studies dealing with both the coverage (e.g., see [12,13,14,15,16]) and interference problems (e.g., see [17,18,19,20]). Closely related to barrier and area coverage, the matching problem is also of interest in the research community (e.g., see [11,21,22,23]).

An important setting in considerations of the coverage of a domain is when the sensors are initially placed at random with a uniform distribution. Some authors proposed using several rounds of random displacement to achieve complete coverage of a domain [24,25]. Another approach is to have the sensors relocate from their initial position to a new position to achieve the desired coverage [26,27].

In this article, we present a novel mathematical theory of the Beta distribution. As an application to sensor networks, we study the most important and difficult cases for the threshold phenomena:On the unit interval when the sensing radius $r_{1}$ is close to $\frac{1}{2n}$ and the interference distance *s* is close to $\frac{1}{n}$, i.e., $r_{1}=\frac{1+\epsilon }{2n}$ and $s=\frac{1-\delta }{n}$;On the unit square when the square sensing radius $r_{2}$ is close to $\frac{1}{2\sqrt{n}}$ and the interference distance *s* is close to $\frac{1}{\sqrt{n}}$, i.e., $r_{2}=\frac{1+\epsilon }{2\lfloor \sqrt{n}\rfloor }$ and $s=\frac{1-\delta }{\left\lfloor \sqrt{n}\right\rfloor }$.
for coverage and interference (see Definition 3), provided that ϵ and 1>δ>0 are arbitrary small constant independent of the number of sensors *n*.

Compared to the coverage problem, the $(r_{m},s)-C\&I$ requirement not only ensures coverage, but also avoids interference and is more reasonable in order to provide reliable communication within the network.

It is worth mentioning that, in this paper, in two dimensions, the sensors can move directly to the final locations with a shortened distance, not only in a vertical and horizontal fashion, as in [28] for the unit square. Hence, our analysis in the current paper when the sensors can move directly to the final locations via the shortest route not only in a vertical and horizontal fashion completes the picture of the threshold phenomena.

More importantly, our investigation is closely related to the papers [28,29] with respect to the analysis of the expected *a*-total displacement for the coverage problem where the sensors are randomly placed on the unit interval [29] and at a higher dimension [28]. Both papers study performance bounds for some algorithms, using Chernoff’s inequality. The methods used in these papers have limitations—the most important and difficult cases when the sensing radius $r_{1}$ is close to $\frac{1}{2n}$ and the square sensing radius $r_{2}$ is close to $\frac{1}{2\sqrt{n}}$ were not included in [28,29]. Moreover, in the paper [28], the sensors can move only parallel to the axes. Hence, the analysis of the coverage problem in [28] is incomplete.

Therefore, it is natural to investigate the general case when the sensor can move directly to the final locations via the shortest route not only in a vertical and horizontal fashion.

Finally, it is worth mentioning that our work is related to the series of papers [6,30,31,32]. In [6,31], the author investigated the maximum of the expected sensor’s displacement (the time required) for coverage and interference. In [6,31], it was assumed that the *n* sensors are initially deployed on [0,∞) according to the arrival times of the Poisson process with arrival rate λ>0, and coverage (connectivity) is in the sense that there are no uncovered points from the origin to the last rightmost sensor. The work by [30] investigated the expected minimal *a*-total displacement for the interference–connectivity requirement when the *n* sensors are initially placed on ${\left[0,\infty \right)}^{d}$ according to *d* identical and independent Poisson processes, each with arrival rate λ>0. It is worth pointing out that the *d*-dimensional model in [30] is only the direct extension of the interference–connectivity requirement from one dimension to the *d*-dimensional space and the sensors move only parallel to the axes (see Table 3).

## 2. Results on the Beta Distribution

In this section, we provide three lemmas about the Beta distribution pertinent for the $\left(r_{m},s\right)$-C&I requirement in WSNs. We also introduce some basic concepts and notations that will be used in the sequel.

In this paper, in the one-dimensional scenario, the *n* mobile sensors are thrown independently at random following a uniform distribution in the unit interval [0,1]. Let $X_{\left(\ell \right)}$ be the position of the *ℓ*-th sensor after sorting the initial random locations of *n* sensors with respect to the origin of the interval [0,1], i.e., the *ℓ*-th-order statistics of the uniform distribution in the unit interval. It is known that the random variable $X_{\left(\ell \right)}$ obeys the Beta distribution with parameters ℓ,n+1−ℓ (see [33], p. 13).

Assume that c,d are positive integers. The Beta distribution Beta(c,d) (see [34]) with parameters c,d is the continuous distribution on [0,1] with the probability density function $f_{c,d}\left(t\right)$ given by
(3)fc,d(t)=cc+d−1ctc−1(1−t)d−1,when0≤t≤1.

The cumulative distribution function of the Beta distribution with parameters c,d is given by the incomplete Beta function:(4)Iz(c,d)=cc+d−1c∫0ztc−1(1−t)d−1dtfor0≤z≤1.

Moreover, the incomplete Beta function is related to the binomial distribution by
(5)1−Iz(c,d)=∑j=0c−1c+d−1jzj(1−z)c+d−1−j
(see [34], Identity 8.17.5, for c:=m,d:=n−m+1, and x:=z) and the binomial identity:(6)∑j=0c+d−1c+d−1jzj(1−z)c+d−1−j=1.

The following inequality, which relates the binomial and Poisson distribution, was discovered by Yu. V. Prohorov (see [35], Theorem 2; [36]).
(7)njxj(1−x)n−j≤nm112e−nx(nx)jj!,
where $m_{1}$ is some integer that satisfies $n\left(1-x\right)-1<m_{1}\leq n\left(1-x\right)$.

We also use the classical Stirling’s approximation for the factorial (see [37], p. 54):(8)$$\sqrt{2\pi }N^{N+\frac{1}{2}}e^{-N+\frac{1}{12N+1}}<N!<\sqrt{2\pi }N^{N+\frac{1}{2}}e^{-N+\frac{1}{12N}}.$$

We use the following notation ${\left|x\right|}^{+}=\max\left\{x,0\right\}$ for the positive parts of x∈R.

We are now ready to give some useful properties of the Beta distribution in the following sequences of lemmas.

**Lemma** **1.**
*Let a>0. Assume that n is a positive integer. Then,*

$$\mathrm{Pr}\left[\mathrm{Beta}\left(n,1\right)<1-\frac{1}{n^{\frac{a}{1+a}}}\right]<\frac{1}{e^{n^{\frac{1}{1+a}}}}.$$



**Lemma** **2.**
*Let a>0 be a constant. Fix γ>0 independently of n. Let $\rho =\frac{1+\gamma }{n}$. Assume that ℓ,n are positive integers and ℓ≤n. Then,*

(9)
$$E\left[{\left({\left|\mathrm{Beta}\left(\ell ,n-\ell +1\right)-\rho \ell \right|}^{+}\right)}^{a}\right]=O\left(\frac{1}{n^{a}}\right),\;\;\;\mathrm{uniformly}\;\mathrm{in}\;\;\;\ell \in \left\{1,2,\ldots ,n\right\},$$


(10)
$$\sum_{\ell =1}^{n}\frac{n}{\ell }E\left[{\left({\left|\mathrm{Beta}\left(\ell ,n-\ell +1\right)-\rho \ell \right|}^{+}\right)}^{a}\right]=O\left(n^{1-a}\right).$$



**Lemma** **3.**
*Let a>0 be a constant. Fix 1>δ>0 independently of n. Let $s=\frac{1-\delta }{n}$. Assume that ℓ,n are positive integers and ℓ≤n. Then,*

(11)
$$\sum_{\ell =1}^{n}\frac{n}{\ell }E\left[{\left({\left|s\ell -\mathrm{Beta}\left(\ell ,n-\ell +1\right)\right|}^{+}\right)}^{a}\right]=O\left(n^{1-a}\right).$$



The following lemma will simplify the upper bound estimations in Section 5 and Section 6.

**Lemma** **4.**
*Fix a>0. Assume that the sensor movement M is the finite sum of movements $M_{i}$ for i=1, 2, ⋯, l, i.e., $M=\sum_{i=1}^{\ell }M_{i}$. Then,*

$$E\left[M^{a}\right]\leq C_{a,\ell }\sum_{i=1}^{\ell }E\left[M_{i}^{a}\right],$$

*where $C_{a,\ell }$ is some constant, which depends only on fixed a and ℓ.*


## 3. Coverage and Interference Requirement When the Sensing Radius $r_{1}=\frac{1}{2n}$ and the Interference Distance $s=\frac{1}{n}$

In this section, we recall the known results about the expected *a*-total displacement to fulfil the $(r_{1},s)-C\&I$ requirement when *n* mobile sensors with the identical sensing radius $r_{1}=\frac{1}{2n}$ are distributed uniformly at random and independently on the unit interval [0,1]. That is, the sum of the sensing area of *n* sensors is equal to the length of the unit interval. Observe that, in the case when the sensing radius $r_{1}=\frac{1}{2n}$ and the interference distance $s=\frac{1}{n}$, the only way to achieve the $\left(r_{1},s\right)$-coverage and interference requirement on the unit interval [0,1] is for the sensors to occupy the equidistant anchor positions $\frac{i}{n}-\frac{1}{2n}$, for i=1, 2, …, n (see Figure 3a). The following exact asymptotic result was proven in [29].

**Theorem** **1**([29])**.**
*Let a be an even positive natural number. Assume that n mobile sensors are thrown uniformly and independently at random on the unit interval [0,1]. The expected a-total displacement of all n sensors when the i-th sensor is sorted in increasing order moves from its current random location to the equidistant anchor location $\frac{i}{n}-\frac{1}{2n}$, for i=1,2,…,n, respectively, is $\frac{\left(\frac{a}{2}\right)!}{2^{\frac{a}{2}}\left(1+a\right)}n^{1-\frac{a}{2}}+O\left(n^{-\frac{a}{2}}\right)$.*

In [10], Theorem 1 was extended to all real-valued exponents a>0.

**Theorem** **2** ([10])**.**
*Fix a>0. Assume that n mobile sensors are thrown uniformly and independently at random on the unit interval [0,1]. The expected a-total displacement of all n sensors, when the i-th sensor sorted in increasing order moves from its current random location to the equidistant anchor location $\frac{i}{n}-\frac{1}{2n}$, for i=1,2,…,n, respectively, is*
(12)$$\frac{\Gamma \left(\frac{a}{2}+1\right)}{2^{\frac{a}{2}}\left(1+a\right)}n^{1-\frac{a}{2}}+O\left(n^{-\frac{a}{2}}\right).$$

The gamma function Γ(a) is defined to be an extension of the factorial to real number arguments. It is related to the factorial by $\Gamma \left(\frac{a}{2}+1\right)=\left(\frac{a}{2}\right)!$ provided that $\frac{a}{2}\in N$. It is also worthwhile to mention that the extension of the direct combinatorial method from [29] leads to the exact asymptotic result in Theorem 2 only when *a* is an odd natural number (see [38], Theorem 2).

## 4. Coverage and Interference Requirement When the Square Sensing Radius $r_{2}=\frac{1}{2\sqrt{n}}$ and the Interference Distance $s=\frac{1}{\sqrt{n}}$

In this section, we analyse the expected *a*-total displacement to achieve the $(r_{2},s)-C\&I$ requirement when *n* mobile sensors with the identical square sensing radius $r_{2}=\frac{1}{2\sqrt{n}}$ are thrown uniformly at random and independently on the unit square ${\left[0,1\right]}^{2},$ provided that *n* is the square of a natural number. That is, the sum of the sensing area of *n* sensors is equal to the area of the unit square.

Observe that, to fulfil the $\left(\frac{1}{\sqrt{n}},\frac{1}{2\sqrt{n}}\right)$-coverage and interference requirement, the sensors have to occupy the following anchor positions $\left(\frac{k}{\sqrt{n}}-\frac{1}{2\sqrt{n}},\frac{l}{\sqrt{n}}-\frac{1}{2\sqrt{n}}\right),$ where $1\leq k,l\leq \sqrt{n}$ and *n* must be the square of a natural number (see Figure 3b).

It is known that the expected one-total displacement in this case is $\Theta \left(\sqrt{\ln(n)n}\right).$ Namely, the following theorem about the optimal transportation cost for random matching was obtained in [11], a book related to these problems, which developed modern methods to bound stochastic processes.

**Theorem** **3**([11], Chapter 4.3)**.**
*Let $n=q^{2}$ for some q∈N. Assume that n mobile sensors $X_{1},\;X_{2},\;\ldots ,\;X_{n}$ are thrown uniformly and independently at random on the unit square ${\left[0,1\right]}^{2}.$ Consider the non-random points ${\left(Z_{i}\right)}_{i\leq n}$ evenly distributed as follows: $Z_{i}=\left(\frac{k}{\sqrt{n}}-\frac{1}{2\sqrt{n}},\frac{l}{\sqrt{n}}-\frac{1}{2\sqrt{n}}\right),$ where $1\leq k,l\leq \sqrt{n},\;\;$$i=k\sqrt{n}+l.$ Then,*
$$E\left(\min_{\pi }\sum_{i=1}^{n}d\left(X_{i},Z_{\pi (i)}\right)\right)=\Theta \left(\sqrt{\ln(n)n}\right),$$
*where the infimum is over all permutations of {1,2,…,n} and where d is the Euclidean distance.*

We are now ready to extend Theorem 3 to the displacement to the power *a* provided that a>1.

**Theorem** **4.**
*Fix a>1. Let $n=q^{2}$ for some q∈N. Assume that n mobile sensors $X_{1},X_{2},\ldots ,X_{n}$ are thrown uniformly and independently at random on the unit square ${\left[0,1\right]}^{2}.$ Consider the non-random points ${\left(Z_{i}\right)}_{i\leq n}$ evenly distributed as follows:*

*$Z_{i}=\left(\frac{k}{\sqrt{n}}-\frac{1}{2\sqrt{n}},\frac{l}{\sqrt{n}}-\frac{1}{2\sqrt{n}}\right),$ where $1\leq k,l\leq \sqrt{n},\;\;$$i=k\sqrt{n}+l.$ Then,*

$$E\left(\min_{\pi }\sum_{i=1}^{n}d^{a}\left(X_{i},Z_{\pi (i)}\right)\right)=\Omega \left({\left(\ln\left(n\right)\right)}^{\frac{a}{2}}n^{1-\frac{a}{2}}\right),$$

*where the infimum is over all permutations of {1,2,…,n} and where d is the Euclidean distance.*


## 5. Coverage and Interference Requirement When the Sensing Radius $r_{1}>\frac{1}{2n}$ and the Interference Distance $s<\frac{1}{n}$

In this section, we analyse the expected *a*-total displacement to fulfil the $(r_{1},s)-C\&I$ requirement when *n* mobile sensors with the identical sensing radius $r_{1}>\frac{1}{2n}$ are distributed uniformly at random and independently on the unit interval [0,1]. That is, the sum of sensing area of *n* sensors is greater than the length of the unit interval.

### 5.1. Analysis of Algorithm 1

Fix a>0. Let γ>0 and 1>δ>0 be arbitrary small constants independent of the number of sensors *n*, and let $\rho =\frac{1+\gamma }{n},\;\;$$s=\frac{1-\delta }{n}.$

This subsection is concerned with reallocating the *n* random sensors within the unit interval to achieve only the following property:The distance between consecutive sensors is greater than or equal to *s* and less than or equal to ρ.The first leftmost sensor is at a distance less than or equal to $\frac{\rho }{2}$ from the origin.

We present a basic and energy-efficient algorithm MV(n,ρ,s) (see Algorithm 1). To illustrate Algorithm 1, let us consider the following simple example. We consider the initial location $0\leq X_{\left(1\right)}\leq X_{\left(2\right)}\leq X_{\left(3\right)}\leq X_{\left(4\right)}\leq 1$ of four sensors on the unit interval such that $X_{\left(1\right)}=\rho ,$$X_{\left(2\right)}=\frac{3}{2}\rho ,$$X_{\left(3\right)}=\frac{3}{2}\rho +\frac{1}{2}s,$$X_{\left(4\right)}=\frac{3}{2}\rho +\frac{3}{4}s$ (see Figure 4).

Firstly, Algorithm 1 moves the sensor $X_{\left(1\right)}$ right-to-left at the position $\frac{\rho }{2},$ and the sensor $X_{\left(2\right)}$ does not move. Then, Algorithm 1 moves the sensor $X_{\left(3\right)}$ left-to-right at the position $\frac{3}{2}\rho +s,$ and the sensor $X_{\left(4\right)}$ left-to-right at the position $\frac{3}{2}\rho +2s.$

Theorem 5 states that the expected *a*-total displacement of algorithm MV(n,ρ,s) is in $O\left(n^{1-a}\right)$ when $\rho =\frac{1+\gamma }{n}$ and $s=\frac{1-\delta }{n}.$ Algorithm 1 is very simple, but the asymptotic analysis is not totally trivial. We note that the asymptotic analysis of Algorithm 1 is crucial in deriving the threshold phenomena.

In the proof of Theorem 5, we combine combinatorial techniques with the properties of the Beta distribution (see Equation (Equation 10) in Lemma 2 and Equation (Equation 11) in Lemma 3). The estimations for the Beta distribution with special positive integer parameters in Lemma 2 and Lemma 3 are new to the best of the author’s knowledge.

We now briefly discuss one technical issue in Steps (3)–(4) of Algorithm 1. It may happen that, for some initial random location of *n* sensors $X_{\left(1\right)}\leq X_{\left(2\right)}\leq \cdots \leq X_{\left(n\right)}$, Algorithm 1 moves some sensors to the right endpoint of the interval [0,1]. Namely, there exists $l_{0}\in N_{+}$ with the following property: $X_{\left(i\right)}$ moves to some point in [0,1) for all $i=1,2,\ldots ,l_{0}$, and $X_{\left(i\right)}$ moves to the right endpoint of the interval [0,1] for all $i=l_{0}+1,\;l_{0}+2,\;\ldots ,\;n.$ Let $Y_{1},\;Y_{2},\;\ldots ,\;Y_{n}$ be the location of *n* sensors $X_{\left(1\right)}\leq X_{\left(2\right)}\leq \cdots \leq X_{\left(n\right)}$ after Algorithm 1. Then, to avoid interference to achieve the property that the distance between consecutive sensors is greater than or equal to *s*, we have to deactivate some sensors. Namely:If $1-Y_{l_{0}}<s$, then for all $i=l_{0}+1,\;l_{0}+2,\;\ldots ,\;n$, the sensors $X_{\left(i\right)}$ will no longer sense;If $1-Y_{l_{0}}\geq s,$ then for all $i=l_{0}+2,\;l_{0}+3,\;\ldots ,\;n$, the sensors $X_{\left(i\right)}$ will no longer sense.

**Theorem** **5.**
*Let a>0 be a constant. Fix γ>0 and 1>δ>0 independently of the number of sensors n. Assume that n mobile sensors are thrown uniformly and independently at random on the unit interval [0,1]. Then, Algorithm 1 for $\rho =\frac{1+\gamma }{n}$ and $s=\frac{1-\delta }{n}$ reallocates the random sensors within the unit interval so that:*
(*i*) 
*The distance between consecutive sensors is greater than or equal to s and less than or equal to ρ.*
(*ii*) 
*The leftmost sensor is at a distance less than or equal to $\frac{\rho }{2}$ from the origin.*
(*iii*) 
*The expected a-total displacement is $O\left(n^{1-a}\right).$*



Notice that Theorem 5 is valid regardless of the sensing radius; it depends only on the fact that the relocated sensors are not too far.

Finally, the following lemma will be helpful in the proof of the main results in Section 5.2 for the sensors on the unit interval. In the proof of Lemma 5, we combine probabilistic techniques together with Estimation (Equation 9) in Lemma 2 for the Beta distribution from Section 2.

**Lemma** **5.**
*Let a>0 be a constant. Fix γ>0 and 1>δ>0 independently of the number of sensors n. Let $\rho =\frac{1+\gamma }{n}$ and $s=\frac{1-\delta }{n}.$ Let $Y_{n}$ be the location of the n-th sensor after algorithm MV(n,ρ,s). Then,*

$$\mathrm{Pr}\left[Y_{n}<1-\frac{2}{n^{\frac{a}{a+1}}}\right]=O\left(\frac{1}{n^{\frac{a}{2}}}\right).$$



### 5.2. Analysis of Algorithm 2

Let us recall that a>0 is fixed and ϵ>0 and 1>δ>0 are arbitrary small constants independent of the number of sensors n. In this subsection, we present algorithm $CV_{1}\left(n,r_{1},s\right)$ (see Algorithm 2) for the $(r_{1},s)-C\&I$ requirement. We prove that the expected *a*-total displacement of algorithm $CV_{1}\left(n,r,s\right)$ is in $O\left(n^{1-a}\right)$ when $r_{1}=\frac{1+\epsilon }{2n}$ and $s=\frac{1-\delta }{n}.$ Notice that our Algorithm 2 consists of two phases. During the first phase (see Initialization), we apply Algorithm 1. Then, in the second phase (see Case B and Case C), we add the additional sensors’ movement. Let $Y_{n}$ be the location of sensors $X_{\left(n\right)}$ after Algorithm 2. The additional movement depends on the position of sensor $Y_{n}$ in the interval [0,1].

We now briefly explain the ideas behind the proof of Theorem 6 and the correctness of Algorithm 2:(i)We have initially *n* random sensors on the unit interval with the identical sensing radius $r_{1}=\frac{1+\epsilon }{2n}.$ Firstly, we apply Algorithm 1 for $\rho =\frac{1+\frac{\epsilon }{2}}{n}$ and $s=\frac{1-\delta }{n}$ to achieve only the following properties:
-The distance between consecutive sensors is greater than or equal to $\frac{1-\delta }{n}$ and less than or equal to $\frac{1+\frac{\epsilon }{2}}{n}.$-The first leftmost sensor is at a distance less than or equal to $\frac{1+\frac{\epsilon }{2}}{2n}$ from the origin.
Applying Theorem 5, we deduce that the expected *a*-total displacement in the Initialization of Algorithm 2 is $O\left(n^{1-a}\right).$(ii)Since the sensors have sensing radius $r_{1}=\frac{1+\epsilon }{2n}$ and the distance between consecutive sensors is less than or equal to $\frac{1+\frac{\epsilon }{2}}{n}=2r_{1}-\frac{\frac{\epsilon }{2}}{n},$ the $\left(r_{1},s\right)$-coverage and interference requirement is solved in $O\left(n^{1-a}\right)$ expected *a*-total displacement in Case B of Algorithm 2. In this case, only a fraction of $\Theta \left(n^{\frac{1}{a+1}}\right)$ of rightmost sensors can move. We upper-bound the movement to the power *a* of each these sensors by $\frac{2^{a}}{n^{\frac{a^{2}}{a+1}}}$ (see Case 2 in the proof of Theorem 6).(iii)In Case C, we move the sensors to equidistant anchor locations in $\Theta \left(n^{1-\frac{a}{2}}\right)$ expected *a*-total displacement. However, we can upper-bound the probability with which Case C occurs (see Lemma 5) to achieve the desired $O\left(n^{1-a}\right)$ expected *a*-total displacement.

We are now ready to prove the main theorem for the sensors on the unit interval.

**Theorem** **6.**
*Let a>0 be a constant. Fix ϵ>0 and 1>δ>0 independently of the number of sensors n. Let $s=\frac{1-\delta }{n}.$ Assume that n mobile sensors with the identical sensing radius $r_{1}=\frac{1+\epsilon }{2n}$ are thrown uniformly and independently at random on the unit interval [0,1]. Then, Algorithm 2 solves the $\left(r_{1},s\right)$-coverage and interference requirement and has expected a-total displacement $O\left(n^{1-a}\right).$*


**Proof.** There are three cases to consider:Case 1: The algorithm terminates after Step 3. This case adds nothing to the expected *a*-total displacement.Case 2: The algorithm terminates after Step 8. Then, $Y_{n}\in \left(1-\frac{2}{n^{\frac{a}{a+1}}},1-r\right).$Let us recall that $r_{1}=\frac{1+\epsilon }{2n},\;\;\;\rho =\frac{1+\frac{\epsilon }{2}}{n}$ and the distance between consecutive sensors is less than or equal to ρ. Hence, we upper-bound the movement to the power *a* of the (n−i)-th sensor for i≥1 as follows:
$$\begin{array}{cc} \; & {\left({\left|1-r_{1}-\left(n-i\right)2r_{1}-\left(1-\frac{2}{n^{\frac{a}{a+1}}}-\rho \left(n-i\right)\right)\right|}^{+}\right)}^{a}\\ \; & ={\left({\left|\frac{2}{n^{\frac{a}{a+1}}}-\frac{\epsilon (n-i)+1+\epsilon }{2n}\right|}^{+}\right)}^{a}\leq \frac{2^{a}}{n^{\frac{a^{2}}{a+1}}}. \end{array}$$
Observe that the movement of the (n−i)-th sensor is positive only when
$$n-i\leq \frac{4n^{\frac{1}{a+1}}}{\epsilon }-\frac{1}{\epsilon }=\Theta \left(n^{\frac{1}{a+1}}\right).$$
From this, we see that only $\Theta \left(n^{\frac{1}{a+1}}\right)$ sensors can move.Observe that the movement to the power *a* of the *n*-th sensor is also less than $\frac{2^{a}}{n^{\frac{a^{2}}{a+1}}}.$Hence, this adds to the *a*-total displacement:
$$\frac{2^{a}}{n^{\frac{a^{2}}{a+1}}}\left(\Theta \left(n^{\frac{1}{a+1}}\right)+1\right)=O\left(n^{1-a}\right).$$Case 3: The algorithm terminates after Step 12. Then, $Y_{n}\leq 1-\frac{2}{n^{\frac{a}{a+1}}}.$In this case, we upper-bound the expected *a*-total displacement in Steps (5)–(7) of algorithm $CV_{1}\left(n,r_{1},s\right)$ by $O\left(n^{1-\frac{a}{2}}\right).$ Then, by Lemma 5, the probability that this case can occur is $O\left(\frac{1}{n^{\frac{a}{2}}}\right)$, and this adds to the expected *a*-total displacement at most:
$$O\left(n^{1-\frac{a}{2}}\right)O\left(\frac{1}{n^{\frac{a}{2}}}\right)=O\left(n^{1-a}\right).$$Finally, combining together the estimation from the Initialization (see Theorem 5), Case 1, Case 2, Case 3, as well as Lemma 4, we conclude that the expected *a*-total displacement of algorithm $CV_{1}\left(n,s,r\right)$ is at most $O\left(n^{1-a}\right).$ This is enough to prove Theorem 6.    □

## 6. Coverage and Interference Requirement for Square Sensing Radius $r_{2}>\frac{1}{2\sqrt{n}}$ and Interference Distance $s<\frac{1}{\sqrt{n}}$

In this section, we analyse the expected *a*-total displacement to achieve the $(r_{2},s)-C\&I$ requirement when *n* mobile sensors with the identical square sensing radius $r_{2}>\frac{1}{2\sqrt{n}}$ are thrown uniformly at random and independently on the unit square ${\left[0,1\right]}^{2},$ That is, the sum of the sensing area of *n* sensors is greater than the area of the unit square.

Let us recall that a>0 is constant and ϵ,δ>0 are fixed arbitrary small constant independent of the number of sensors n.

We prove that the expected *a*-total expected displacement of the algorithm $CV_{2}\left(n,r_{2},s\right)$ (see Algorithm 3) is in $O\left(n^{1-\frac{a}{2}}\right)$ when $r_{2}=\frac{1+\epsilon }{2\lfloor \sqrt{n}\rfloor }$ and $s=\frac{1-\delta }{\left\lfloor \sqrt{n}\right\rfloor }.$

Notice that our Algorithm 3 is in two phases. During the first phase (see Steps (1)–(7)), we use a greedy strategy and move all the sensors only according to the second coordinate. As a result of the first phase, we obtain $\left\lfloor \sqrt{n}\right\rfloor$ lines, each with $\left\lfloor \sqrt{n}\right\rfloor$ random sensors. For the second phase, the main result from Section 5 (see Theorem 6) is applicable.

It is worth pointing out that the first phase of Algorithm 3 reduces the *a*-total displacement on the unit square to the *a*-total displacement on the unit interval. Obviously, Algorithm 3 moves sensors only in a vertical and horizontal fashion, but it is powerful enough to derive the desired threshold.

We are now ready to prove the main result for the sensor on the unit square.

**Theorem** **7.**
*Let a>0 be a constant. Fix ϵ>0 and 1>δ>0 as arbitrary small constants independently of the number of sensors n. Let $s=\frac{1-\delta }{\left\lfloor \sqrt{n}\right\rfloor }.$ Assume that n mobile sensors with the identical square sensing radius $r_{2}=\frac{1+\epsilon }{2\lfloor \sqrt{n}\rfloor }$ are thrown uniformly and independently at random on the unit square ${\left[0,1\right]}^{2}.$ Then, Algorithm 3 solves the $\left(r_{2},s\right)$-coverage and interference requirement and has expected a-total displacement in $O\left(n^{1-\frac{a}{2}}\right).$*


**Proof** **of** **Theorem** **7.**Firstly, we look at the expected *a*-total displacement in the first phase of the algorithm (see Steps (1)–(7)). It was proven in [28] that the expected *a*-total displacement in Steps (1)–(7) of Algorithm 3 is in $O\left(n^{1-\frac{a}{2}}\right)$ (see the estimation of $E_{\left(1-6\right)}^{\left(a\right)}$ for $n:={\left(\lfloor \sqrt{n}\rfloor \right)}^{2},$ d=2 in the proof of [28], Theorem 5, Formulas (8) and (10), p. 41).Observe that, in the second phase of Algorithm 3 (see Steps (8)–(10)), we have $\left\lfloor \sqrt{n}\right\rfloor$ lines each with $\left\lfloor \sqrt{n}\right\rfloor$ random sensors with the identical sensing radius $r_{1}=\frac{1+\epsilon }{2\lfloor \sqrt{n}\rfloor }.$ According to Theorem 6, the expected *a*-total displacement is $\left\lfloor \sqrt{n}\right\rfloor O\left({\left(\lfloor \sqrt{n}\rfloor \right)}^{1-a}\right)=O\left(n^{1-\frac{a}{2}}\right).$ This together with Lemma 4 completes the proof of Theorem 7.    □

## 7. Sensors in Higher Dimensions

In this section, we discuss the expected *a*-total displacement for the $\left(r_{m},s\right)$-coverage and interference requirement in higher dimensions, when m>2.

Let us recall that the proposed Algorithm 3 moves the sensors only in a vertical and horizontal fashion and reduces the *a*-total displacement on the unit square to the *a*-total displacement on the unit interval.

Hence, Algorithm 3 can be extended for the random sensors on the *m*-dimensional cube ${\left[0,1\right]}^{m},$ when m>2. We can, similar to the square sensing radius (see Definition 2) define an *m*-dimensional cube sensing radius, move the sensors only according to the axes, and reduce the *a*-total displacement on the unit cube to the *a*-total displacement on the unit interval.

Namely, for the sensors with the identical *m*-cube sensing radius $r_{m}>\frac{1}{2n^{1/m}}$ (the sum of the sensing area of *n* sensors is greater than the area of the unit cube) and the interference distance $s<\frac{1}{n^{1/m}}$, it is possible to give an algorithm with $O\left(n^{1-\frac{a}{m}}\right)$ expected *a*-total displacement for all powers a>0. However, even though Theorem 7 can be generalized for the random sensors with the identical *m*-cube sensing radius $r_{m}>\frac{1}{2n^{1/m}}$ on the *m*-dimensional cube, when m>2, the proposed generalization is weak.

Notice that Theorem 3 is closely related to the main result of paper [21]. Namely, consider two sequences $X_{1},\;X_{2},\;\ldots ,\;X_{n};$
$Y_{1},\;Y_{2},\;\ldots ,\;Y_{n}$ of points that are independently uniformly distributed and the non-random points ${\left(Z_{i}\right)}_{i\leq n}$ are evenly distributed, i.e., $Z_{i}=\left(\frac{k}{\sqrt{n}}-\frac{1}{2\sqrt{n}},\frac{l}{\sqrt{n}}-\frac{1}{2\sqrt{n}}\right),$ where $1\leq k,\;l\leq \sqrt{n},\;\;$$i=k\sqrt{n}+l$ on the unit square ${\left[0,1\right]}^{2}$, then
$$E\left(\underset{\pi }{\inf}\sum_{i=1}^{n}d\left(X_{i},Z_{\pi (i)}\right)\right)=E\left(\underset{\pi }{\inf}\sum_{i=1}^{n}d\left(X_{i},Y_{\pi (i)}\right)\right)=\Theta \left(\sqrt{\ln(n)n}\right),$$
where π ranges over all permutations of {1,2,…,n} and $n=q^{2}$ for some q∈N.

On the other hand, there is a difference between m=2 (the two-dimensional case) and m>2 (the case of dimension at least three). Namely, for two sequences $X_{1},\;X_{2},\;\ldots ,\;$$X_{n};$$Y_{1},Y_{2},\ldots ,Y_{n}$ of points that are independently uniformly distributed on the *m*-dimensional cube ${\left[0,1\right]}^{m},$ when m>2, we have
$$E\left(\underset{\pi }{\inf}\sum_{i=1}^{n}d\left(X_{i},Y_{\pi (i)}\right)\right)=\Theta \left(n^{1-\frac{1}{m}}\right),$$
provided that π ranges over all permutations of {1,2,…,n} (see [39] for details).

Hence, it seems that Theorem 3 together with Theorem 4 can be generalized for *n* random mobile sensors $X_{1},\;X_{2},\;\ldots ,\;X_{n}$ on the *m*-dimensional cube ${\left[0,1\right]}^{m},$ when m>2, and the following result should hold.

Assume that *n* random variables $X_{1},\;X_{2},\;\ldots ,\;X_{n}$ are independently uniformly distributed and the non-random points ${\left(Z_{i}\right)}_{i\leq n}$ evenly distributed at the the positions
$$\left(\frac{l_{1}}{n^{1/d}}-\frac{1}{2n^{1/d}},\frac{l_{2}}{n^{1/d}}-\frac{1}{2n^{1/d}},\ldots ,\frac{l_{d}}{n^{1/d}}-\frac{1}{2n^{1/d}}\right),$$
for $1\leq l_{1},\;l_{2},\;\ldots ,\;l_{d}\leq n^{1/d}$ and $l_{1},\;l_{2},\;\ldots ,\;l_{d}\in N$ on the unit *m*-dimensional cube ${\left[0,1\right]}^{m}$, then
(13)$$E\left(\underset{\pi }{\inf}\sum_{i=1}^{n}d^{a}\left(X_{i},Z_{\pi (i)}\right)\right)=\Theta \left(n^{1-\frac{a}{m}}\right)$$
for all powers a≥1, where π ranges over all permutations of {1,2,…,n} and $n=q^{m}$ for some q∈N.

Therefore, it is an open problem to prove that the $\left(r_{m},s\right)$-coverage and interference requirement for an *m*-cube sensing radius $r_{m}=\frac{1}{2n^{1/m}}$ (the sum of the sensing area of *n* sensors is equal to the area of unit cube) and the interference distance $s=\frac{1}{n^{1/m}}$ can be solved in $\Theta \left(n^{1-\frac{a}{m}}\right)$ and to study the expected *a*-total displacement for the $\left(r_{m},s\right)$-coverage and interference requirement, when $r_{m}>\frac{1}{2n^{1/m}}$ and $s<\frac{1}{n^{1/m}}.$

## 8. Experimental Results

In this section, we provide a set of experiments to confirm the discovered theoretical threshold for the expected *a*-total displacement. Wolfram Mathematica 10.0 was used for our experiments when a=1,
$a=\frac{3}{2}$, and a=2. We distinguish two cases:

Case 1: sensing radius $r_{1}>\frac{1}{2n}$ and interference distance $s<\frac{1}{n}.$

In this case, we conduct Algorithm 4.
**Algorithm 4** Realisation of Algorithm 11:n:=12:**while** n ≤ 5000 **do**3:   Generate independently and uniformly *n* random points on the unit interval [0,1];4:   Calculate $T_{n}^{\left(a\right)}$ according to Algorithm 1 for $\rho =\frac{1.8}{n}$ and $s=\frac{0.5}{n};$5:   Insert the points $\left(n,T_{n}^{\left(a\right)}\right)$ into the chart;6:   n:=n+17:**end while**

Figure 5, Figure 6 and Figure 7 illustrate the described experiment for a=1,
$a=\frac{3}{2}$, and a=2.

Notice that the experimental *a*-total displacement of Algorithm 4 is constant and independent of the number of sensors for a=1, is $\Theta \left(\frac{1}{\sqrt{n}}\right)$ for $a=\frac{3}{2}$, and is $\Theta \left(\frac{1}{n}\right)$ for a=2. Therefore, the carried out experiments confirm very well our theoretical upper bound estimation O(1) for a=1, $O\left(\frac{1}{\sqrt{n}}\right)$ for $a=\frac{3}{2}$, and $O\left(\frac{1}{n}\right)$ for a=2 (see Theorem 5 for a=1,
$a=\frac{3}{2}$, and a=2).

Case 2: sensing radius $r_{1}=\frac{1}{2n}$ and interference distance $s=\frac{1}{n}.$

In this case, we conduct Algorithm 5.
**Algorithm 5** Realisation of Theorem 2 1:n:=1 2:**while** n ≤ 60 **do** 3:   **for** j=1
**to** 200 **do** 4:     Generate independently and uniformly $n^{2}$ random points on the unit interval [0,1]; 5:     Calculate $T_{n^{2}}^{\left(a\right)}\left(j\right)$ according to Theorem 2; 6:   **end for** 7:   **for** k=1
**to** 20 **do** 8:     Calculate the average $T_{n^{2},k}^{\left(a\right)}=\frac{1}{10}\sum_{j=1}^{10}T_{n^{2}}^{\left(a\right)}\left(j+\left(k-1\right)*10\right)$; 9:     Insert the points $\left(n^{2},T_{n^{2},k}^{\left(a\right)}\right)$ into the chart;10:   **end for**11:   n:=n+112:**end while**

In Figure 8, Figure 9 and Figure 10, the black points represent the numerical results of the conducted experiments. The additional lines $\left\{\left(n,\frac{\Gamma \left(\frac{3}{2}\right)}{2\sqrt{2}}\sqrt{n}\right),\;\;1\leq n\leq 3600\right\},$ $\{\left(n,\frac{\Gamma \left(\frac{7}{4}\right)}{2^{\frac{3}{4}}\left(\frac{5}{2}\right)}n^{\frac{1}{4}}\right),\;\;$1≤n≤3600}, $\left\{\left(n,\frac{1}{6}\right),\;\;1\leq n\leq 3600\right\}$ are the plots of a function, which is the theoretical estimation (see the leading term in the asymptotic result of Theorem 2 for a=1,
$a=\frac{3}{2}$, and a=2). It is worth pointing out that numerical results are situated near the theoretical line.

It is also possible to repeat the experiments to all exponents a>0, as well as Algorithms 2 and 3.

## 9. Conclusions and Future Direction

In this paper, the following natural problem was investigated: given *n* uniformly random mobile sensors in an *m*-dimensional unit cube, where m∈{1,2}, what is the minimal energy consumption to move them so that they are pairwise at an interference distance at least *s* apart and so that every point of the *m*-dimensional unit cube is within the range of at least one sensor?

As the energy consumption measure for the displacement of *n* sensors, we considered the *a*-total displacement defined as the sum $\sum_{i=1}^{n}d_{i}^{a},$ where $d_{i}$ is the distance sensor *i* has been moved and a>0. The main findings can be summarized as follows:For the sensors placed on the unit interval, sensing radius $r_{1}=\frac{1}{2n}$, and interference distance $s=\frac{1}{n}$, the expected minimal *a*-total displacement is of order $\Theta \left(n^{1-\frac{a}{2}}\right).$ When $r_{1}=\frac{1+\epsilon }{2n}$ and $s=\frac{1-\delta }{n},$ provided that ϵ>0 and 1>δ>0 are arbitrary small constants independent of the number of sensors n, then there is an algorithm with $O\left(n^{1-a}\right)$ expected *a*-total displacement for all powers a>0.For the case of the unit square and a>0, square sensing radius $r_{2}=\frac{1}{2\sqrt{n}}$, and interference distance $s=\frac{1}{\sqrt{n}}$, the expected minimal *a*-total displacement is at least of order $\Omega \left({\left(\log(n)\right)}^{\frac{a}{2}}n^{1-\frac{a}{2}}\right),$ provided that *n* is the square of a natural number. When $r_{2}=\frac{1+\epsilon }{2\lfloor \sqrt{n}\rfloor }$ and $s=\frac{1-\delta }{\left\lfloor \sqrt{n}\right\rfloor },$ provided that ϵ>0 and 1>δ>0 are arbitrary small constants independent of the number of sensors n, then there is an algorithm with $O\left(n^{1-\frac{a}{2}}\right)$ expected *a*-total displacement for all powers a≥1.

This paper opens several research directions.

First, it would be interesting to know what happens if ϵ and δ depend on *n* and decrease to 0. This would give the complete picture of the threshold phenomena for the coverage and interference requirement.

Second, in this paper, we investigated the coverage and interference requirement only for one- and two-dimensional networks. It is an open problem to generalize this study to higher dimensions and investigate threshold phenomena for the *m*-dimensional cube, similar to 1- and 2-dimensional cubes.

Additionally it would be interesting for future research to study the coverage and interference requirement for a non-uniform displacement of sensors, on other domains, as well for some real-life sensor displacement.

We proved that the energy consumption for the coverage and interference requirement is constant and independent of the number of sensors for some parameters (see Equations (Equation 1) and (Equation 2)). While we discussed the practical importance of this constant energy consumption, an open problem for future study is the experimental evaluation of energy consumption for some real-life sensor displacement. However, this experimental evaluation for some real-life sensor deployment may be rather expensive due to the large number of sensors that would be required.

## Figures and Tables

**Figure 1 sensors-22-08789-f001:**
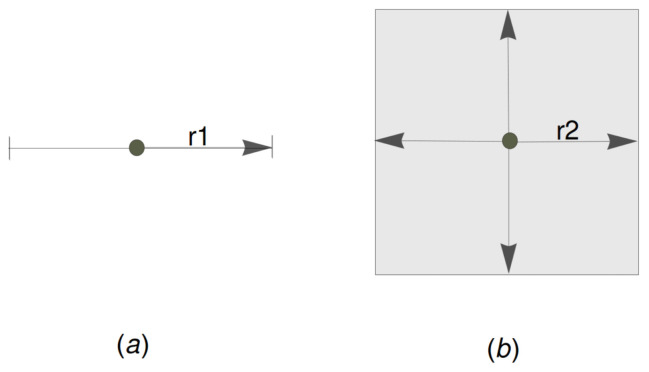
(**a**) Sensing radius $r_{1}$ on a line. (**b**) Square sensing radius $r_{2}$.

**Figure 2 sensors-22-08789-f002:**
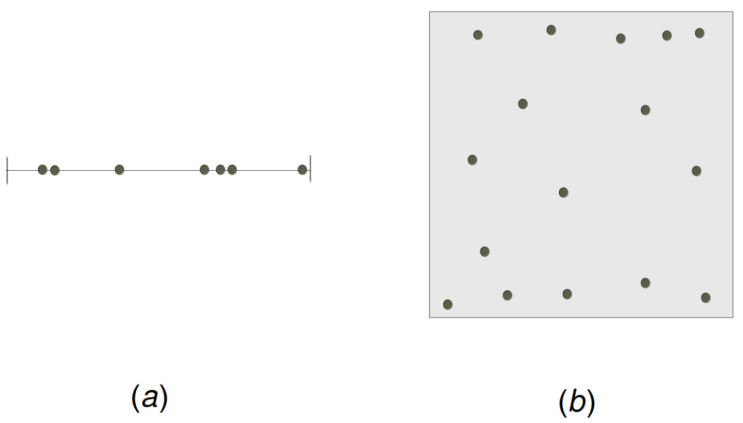
(**a**) Random sensors on the unit interval. (**b**) Random sensors on the unit square.

**Figure 3 sensors-22-08789-f003:**
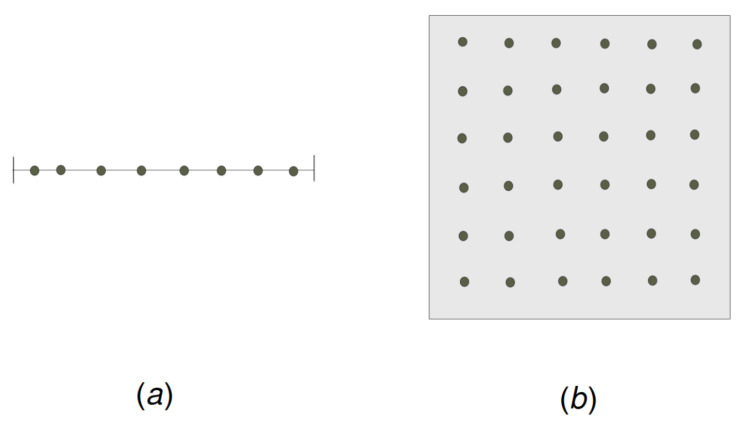
(**a**) Sensors at the anchor positions on the unit interval. (**b**) Sensors at the anchor positions on the unit square.

**Figure 4 sensors-22-08789-f004:**
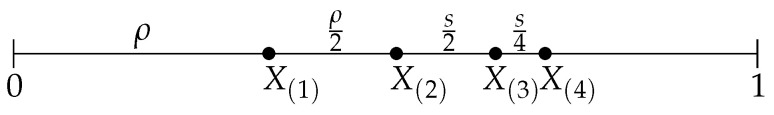
The positions of 4 mobile sensors $X_{\left(1\right)},\;X_{\left(2\right)},\;X_{\left(3\right)},\;X_{\left(4\right)}$ on the unit interval.

**Figure 5 sensors-22-08789-f005:**
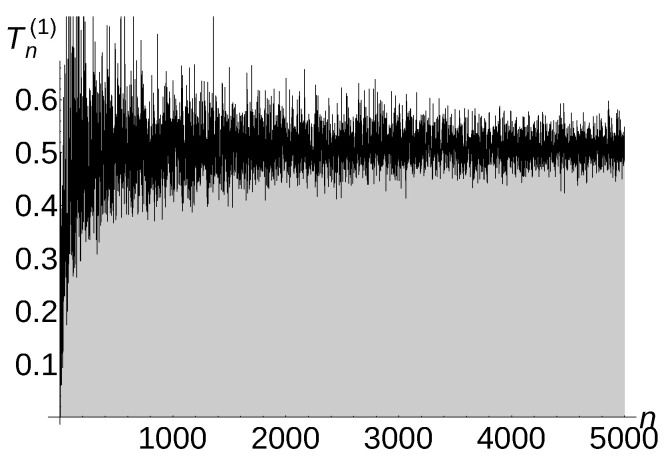
$T_{n}^{\left(1\right)}=\Theta \left(1\right)$ of Algorithm 4.

**Figure 6 sensors-22-08789-f006:**
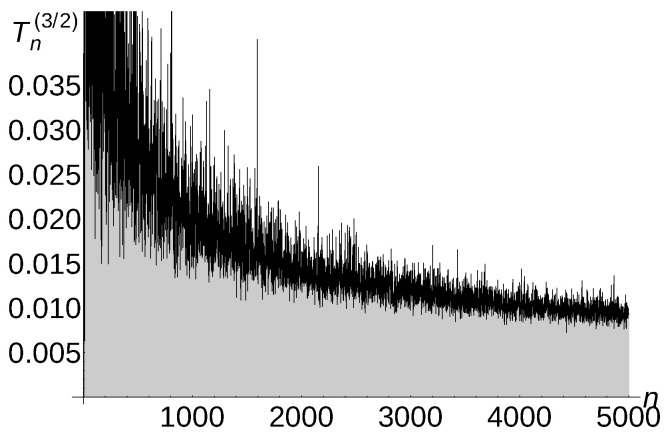
$T_{n}^{\left(3/2\right)}=\Theta \left(\frac{1}{\sqrt{n}}\right)$ of Algorithm 4.

**Figure 7 sensors-22-08789-f007:**
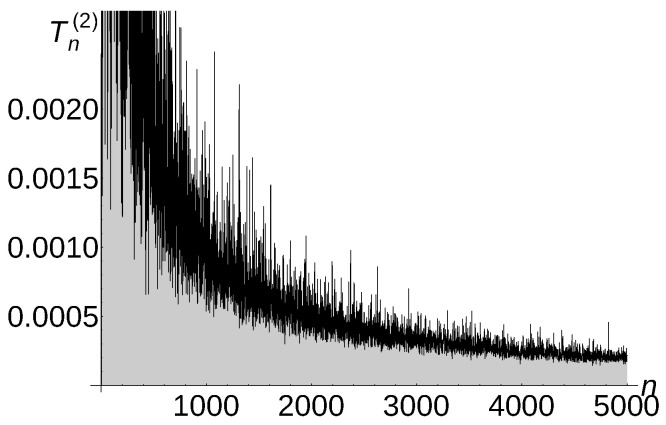
$T_{n}^{\left(2\right)}=\Theta \left(\frac{1}{n}\right)$ of Algorithm 4.

**Figure 8 sensors-22-08789-f008:**
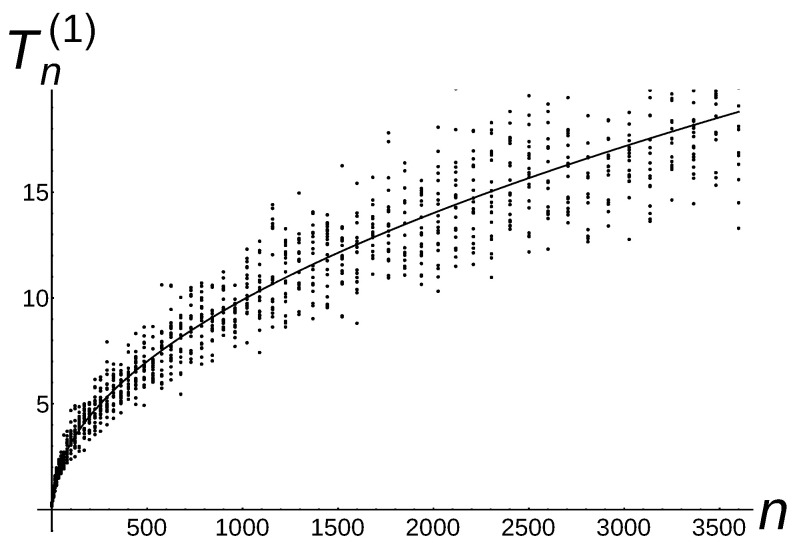
$T_{n}^{\left(1\right)}∼\frac{\Gamma \left(\frac{3}{2}\right)}{2\sqrt{2}}\sqrt{n}$ of Algorithm 5 with the additional theoretical line according to the leading term of Theorem 2 for a=1.

**Figure 9 sensors-22-08789-f009:**
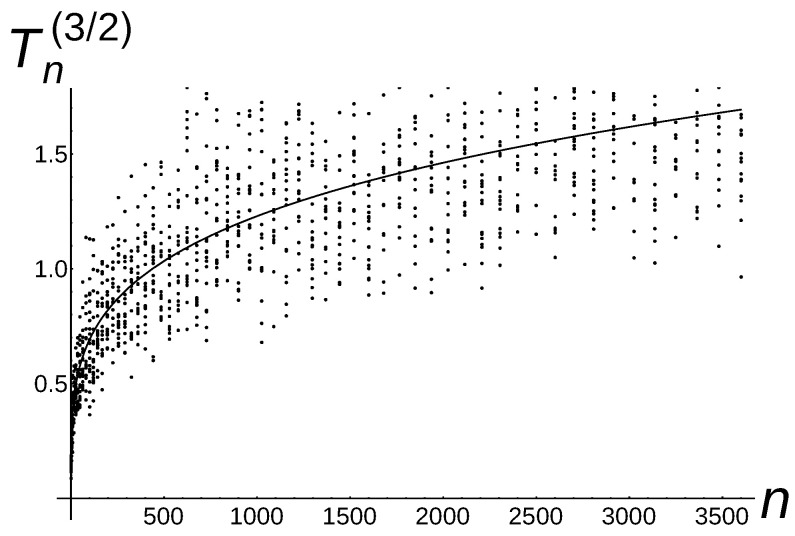
$T_{n}^{\left(3/2\right)}∼\frac{\Gamma \left(\frac{7}{4}\right)}{2^{\frac{3}{4}}\left(\frac{5}{2}\right)}n^{\frac{1}{4}}$ of Algorithm 5 with the additional theoretical line according to the leading term of Theorem 2 for a=3/2.

**Figure 10 sensors-22-08789-f010:**
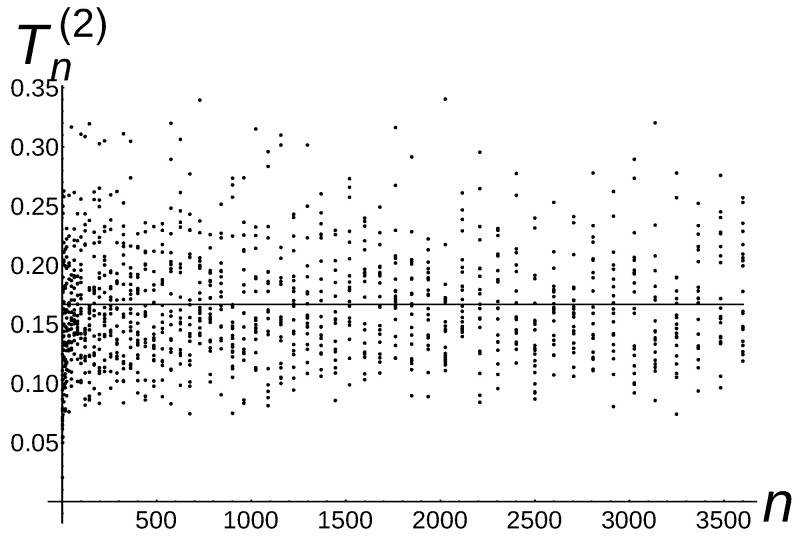
$T_{n}^{\left(2\right)}∼\frac{1}{6}$ of Algorithm 5 with the additional theoretical line according to the leading term of Theorem 2 for a=2.

**Table 1 sensors-22-08789-t001:** The expected minimal *a*-total displacement of *n* random sensors on the unit interval [0,1] as a function of the sensing radius $r_{1}$ and the interference value *s*, where ϵ>0, 1>δ>0.

Sensing Radius $r_{1}$	Interference Distance *s*	Expected Minimal *a*-Total Displacement for $(r_{1},s)-C\&I$ Requirement	Theorem
$r_{1}=\frac{1}{2n}$	$s=\frac{1}{n}$	$\frac{\Gamma (\frac{a}{2}+1)}{2^{\frac{a}{2}}\left(1+a\right)}n^{1-\frac{a}{2}}+O\left(n^{-\frac{a}{2}}\right)$, a>0	Theorem 2 (cf. [10])
$r_{1}=\frac{1+\epsilon }{2n}$, ϵ>0	$s=\frac{1-\delta }{n}$, 1>δ>0	$O\left(n^{1-a}\right),\;\;$ a>0	Theorem 6

**Table 2 sensors-22-08789-t002:** The expected minimal *a*-total displacement of *n* random sensors on the unit square ${\left[0,1\right]}^{2}$ as a function of the square sensing radius $r_{2}$ and the interference value *s*, where ϵ>0, 1>δ>0.

Square Sensing Radius $r_{2}$	Interference Distance *s*	Expected Minimal *a*-Total Displacement for $(r_{2},s)-C\&I$ Requirement	Theorem
$r_{2}=\frac{1}{2\sqrt{n}}$	$s=\frac{1}{\sqrt{n}}$	$\Theta \left(\sqrt{\ln(n)n}\right)$ if a=1 $\Omega \left({\left(\ln\left(n\right)\right)}^{\frac{a}{2}}n^{1-\frac{a}{2}}\right)$ if a>1	Theorem 3 (cf. [11]) Theorem 4
$r_{2}=\frac{1+\epsilon }{2\lfloor \sqrt{n}\rfloor }$, ϵ>0	$s=\frac{1-\delta }{\left\lfloor \sqrt{n}\right\rfloor }$, 1>δ>0	$O\left(n^{1-\frac{a}{2}}\right)$ if a>0	Theorem 7

**Table 3 sensors-22-08789-t003:** Comparison of related papers provided that the symbol ∨ means it is included.

Reference	Deployment Distribution	Energy	1D	Movement in 2D	2D	Requirement
This paper	Random deployment	∨	∨	Direct to the final locations	∨	Coverage and interference
[6]	Poisson process		∨			Coverage and interference
[30]	Poisson process	∨	∨	Only parallel to the axes	∨	Coverage and interference
[18]	Random deployment		∨		∨	Interference
[20]	Random deployment		∨			Interference
[19]	Evenly distributed				∨	Interference
[7]	Poisson process		∨	Only parallel to the axes	∨	Interference
[29]	Random deployment	∨	∨			Coverage
[28]	Random deployment	∨	∨	Only parallel to the axes	∨	Coverage
[31]	Poisson process		∨			Coverage and interference
[1]	Random deployment		∨			Coverage

## Data Availability

Not applicable.

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
