# Peer review of "Analysis of the Threshold for Energy Consumption in Displacement of Random Sensors"

_sensors, 2022, doi:10.3390/s22228789_

Round 1

Reviewer 1 Report

This paper analyzed the energy-efficient reallocation of mobile random sensors, which is a interesting problem. But, it should be improved a lot before publication. My detailed comments are as follows:

1. The novelty of this paper is not sufficient. The interval sensing radius and square sensing radius are investigated many years ago. What difference between the previous study and the research in this paper?Please highlight the motivation, contribution and innovation of the paper.

2. This paper studied the allocation problem for mobile sensors, which is a practical problem. However, there are many mathematical formulation in this paper, which is hard to follow. It is better to use some gragh or other easy-uderstanding stuff to conclude the framework of the paper.

3. The  experimental results is based on the mathematical analysis. Please analyze the energy consumption of the displacement solution.

4. The presentation of this paper should be checked carefully.

Author Response

I would like to thank Reviewer 1 for the very helpful comments. I have revised the paper according to the comments. Below I include the details on the changes made in the paper and my response to the specific comments.

Reviewer 2 Report

Overall, this is a nice paper summarizing the results in the displacement of random sensors on a unit interval or on a unit square. Theoretical results are provided with rigorous proof.

One minor thing to be fixed by the author is to remove the "best paper award" description in lines 47-48. Simply cite the paper [8] would be sufficient.

Author Response

I would like to thank Reviewer 2 for the very helpful comments. I have revised the paper according to the comments. Below I include the details on the changes made in the paper and my response to the specific comments

Reviewer 3 Report

 I would like to thank the authors for their effort in the current study and encourage them to improve the whole manuscript according to the following comments

The abstract can be enhanced where they discuss about the methodology along with achieved results.

The author needs to clearly explain the novelty and motivation of the proposed work.

The author should justify the reason behind choosing random sensors which are used to analyse the threshold for energy consumption.

The architecture for the proposed methodology can be given to understand easily

Recent works related to energy consumption can be referred to and summarised in the table format with drawbacks. Can refer to the load balancing of energy cloud using wind-driven and firefly algorithms in internet of everything

Many equations are derived here and proved using the theorems and definitions. Some attributes not explained clearly. Please check once all

The paper needs proper proofreading and grammar editing.

Author Response

I would like to thank Reviewer 3 for the very helpful comments. I have revised the paper according to the comments. Below I include the details on the changes made in the paper and my response to the specific comments.
